# Towards Unsupervised Open World Semantic Segmentation

**Svenja Uhlemeyer**[1]      **Matthias Rottmann**[1]      **Hanno Gottschalk**[1]

[1]IZMD and Faculty of Mathematics and Natural Sciences, University of Wuppertal, Germany,

## Abstract

For the semantic segmentation of images, state-of-the-art deep neural networks (DNNs) achieve high segmentation accuracy if that task is restricted to a closed set of classes. However, as of now DNNs have limited ability to operate in an open world, where they are tasked to identify pixels belonging to unknown objects and eventually to learn novel classes, incrementally. Humans have the capability to say: "I don't know what that is, but I've already seen something like that". Therefore, it is desirable to perform such an incremental learning task in an unsupervised fashion. We introduce a method where unknown objects are clustered based on visual similarity. Those clusters are utilized to define new classes and serve as training data for unsupervised incremental learning. More precisely, the connected components of a predicted semantic segmentation are assessed by a segmentation quality estimate. Connected components with a low estimated prediction quality are candidates for a subsequent clustering. Additionally, the component-wise quality assessment allows for obtaining predicted segmentation masks for the image regions potentially containing unknown objects. The respective pixels of such masks are pseudo-labeled and afterwards used for re-training the DNN, *i.e.,* without the use of ground truth generated by humans. In our experiments we demonstrate that, without access to ground truth and even with few data, a DNN's class space can be extended by a novel class, achieving considerable segmentation accuracy.

## 1 INTRODUCTION

Semantic segmentation is a computer vision task that terms the classification of image data on pixel level. State-of-the-

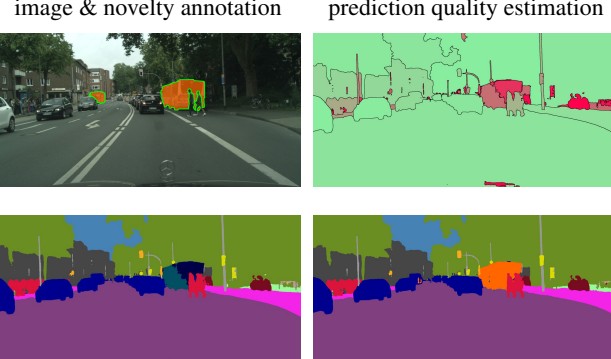

image & novelty annotation     prediction quality estimation

prediction of the initial DNN     prediction of our extended DNN

Figure 1: Comparison of the semantic segmentation predictions of an initial DNN (bottom left) whose semantic space does not include the category *bus* and a DNN which is incrementally extended by this novel class (bottom right, novel class in orange) for an image from the Cityscapes dataset. The novel class is highlighted in orange (top left). Further, the initial prediction exhibits a low prediction quality (top right) on pixels belonging to the novel objects, which is indicated by red color.

art approaches are based on deep convolutional neural networks (DNNs) [Chen et al., 2018b, Wang et al., 2021, Zhao et al., 2017], benefiting from finely annotated datasets, *e.g.,* for automated driving [Cordts et al., 2016, Geyer et al., 2020, Neuhold et al., 2017, Yu et al., 2020]. However, DNNs for semantic segmentation are usually trained on a predefined, closed set of classes. This closed world setting assumes, that all classes present during testing were already included in the training set. In an open world setting, this assumption does not hold. In particular for safety-critical open-world applications like perception systems for automated driving, it is indispensable that neural networks recognize previously unseen objects instead of wrongly assigning them to *one-of-the-known* classes. In addition, they must constantly adapt

*Accepted for the 38th Conference on Uncertainty in Artificial Intelligence* (UAI 2022).

to evolving environments.

Some terms often used interchangeably for anomaly are *outlier*, *out-of-distribution* (OoD) object and *novelty*. As there is no clear convention on how to distinguish these terms, we define them as subcategories of anomalies: outliers and OoD objects denote noise or samples drawn from another distribution than the model was trained on, respectively. In this work, we are seeking novelties, which we define as previously-unseen objects that constitute a new concept, *i.e.,* objects of the same category appear frequently. In automated driving, detecting and learning those novel classes becomes necessary, *e.g.,* due to new appearances like e-scooters or due to local specialities like boat trailers near the sea. The concept of detecting and learning novelties was first introduced in Bendale and Boult [2015] as *open world recognition*. Open world recognition for different computer vision tasks is an emerging research area [Bendale and Boult, 2015, Joseph et al., 2021, Cen et al., 2021, Shu et al., 2018], still only little explored for unsupervised methods [He and Zhu, 2021, Nakajima et al., 2019], yet.

We propose a new and modular procedure for learning new classes of novel objects without any handcrafted annotation:

1. Anomaly segmentation to detect suspicious objects,
2. clustering of potentially novel objects,
3. creation of so-called *pseudo labels*, and
4. incremental learning of novel classes.

In the following, we will outline each of these four steps in more detail.

For the first step, we post-process the predictions of an underlying semantic segmentation DNN via a *meta regressor*, that estimates the quality of the predicted segments, similar as proposed in Rottmann and Schubert [2019], Rottmann et al. [2020], Maag et al. [2020]. In the following, the term **segment** will always refer to connected components of pixels in the semantic segmentation prediction. The segment-wise quality score is obtained on the basis of aggregated dispersion measures and geometrical information, *i.e.,* without requiring ground truth. The output of the semantic segmentation DNN on anomalous objects is often split into several segments. To this end, we first aggregate neighboring segments, *i.e.,* segments that have at least one adjacent pixel each, with quality estimates below some threshold, into (potentially) anomalous objects, termed **suspicious objects**.

For the second step, we adapt the idea introduced in Oberdiek et al. [2020] to gather segments with poor prediction quality and to cluster them into visually related neighborhoods. Therefore, all suspicious objects (of sufficient size) are cropped out in the RGB images and the resulting image patches are fed into a convolutional neural network (CNN), *e.g.,* for image classification. Whether an image patch is sufficiently large depends on the minimum input size required by this CNN. To obtain comparable information about the suspicious objects, we then extract the features provided by the penultimate layer of the CNN, *i.e.,* right before the final classification layer. By reducing the dimensionality of these features up to two, we enable the use of low-dimensional, unsupervised clustering techniques, such as Ester et al. [1996], MacQueen [1967].

As third, we obtain pseudo labels for novel classes in an automated manner: each (large / dense enough) cluster constitutes a novel category, and each pixel belonging to a clustered object is assigned to the appropriate (not necessarily named) class. More precisely, the prediction of the segmentation model is updated at those pixel positions to the next "free" label ID.

Finally, the segmentation network is incrementally extended by these novel classes (see Fig. 1 for an example). To this end, we apply established incremental learning methods [Hinton et al., 2015, Robins, 1995]. However, these are mainly examined for supervised learning tasks, while we do not include any hand-labeled new data. This last two steps were never done in literature so far.

We perform five experiments, following a hierarchical structure of complexity. For the first three experiments, the initial segmentation network is trained on the Cityscapes dataset, but on different subsets of the available training classes. Here, we do not change the data itself, but the training IDs of the Cityscapes classes. For the other experiments, we start with an initial segmentation network that is trained on Cityscapes and test our method on the A2D2 dataset. For those, we have a mapping between the Cityscapes and the A2D2 classes. For most Cityscapes classes, there is a matching class in A2D2. In some cases, A2D2 has coarser classes, *e.g.,* we map the Cityscapes classes *vegetation* and *terrain* to the A2D2 class *nature*.

To outline our contributions, we demonstrate in our experiments that our method is able to incrementally extend a neural network by novel classes without collecting or annotating novelties manually. To the best of our knowledge, we are the first to introduce an unsupervised approach for open world semantic segmentation with DNNs. Fine-tuning neural networks on automatically created pseudo-labels instead of human-made annotations is economically valuable. We observe in all experiments, that even a poor labeling quality is sufficient to learn novel classes, achieving IoU values around $40\%$. Further, the amount of new data was less mostly than 100 images, respectively. Unsupervised open world semantic segmentation therefore is a powerful tool for open world applications, that provides an enormous potential for future improvement.

## 2 RELATED WORK

In this section, we first review anomaly detection methods and briefly go into class discovery approaches. Then we

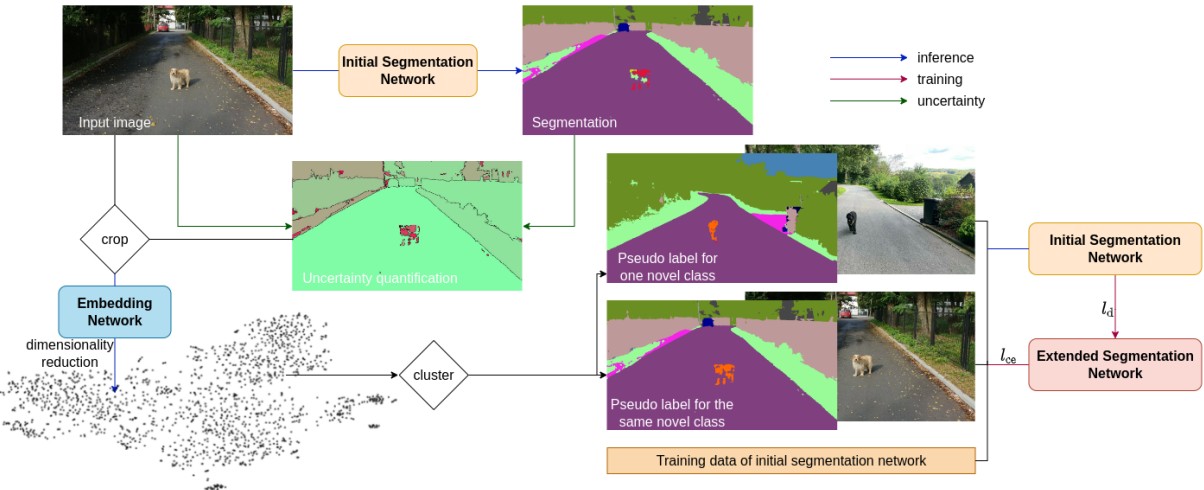

Figure 2: Illustration of the overall framework.

describe different strategies for class-incremental learning. Finally, we give an overview of existing work on open world computer vision tasks.

**Novelty Detection.**   The detection of anomalous objects in general is a key task in many machine learning applications. Early works estimate the prediction uncertainty, *e.g.,* by uncertainty measures derived from the softmax probability [Hendrycks and Gimpel, 2017, Liang et al., 2018]. Uncertainty-based approaches can be further improved by integrating anomalous data into the training procedure [Devries and Taylor, 2018, Chan et al., 2021b]. Another line of works employs generative models such as autoencoders (AEs) or generative adversarial models (GANs) to reconstruct or synthesise images and measure the reconstruction quality. Various of those novelty detection methods are described in Vasilev et al. [2018], not only reconstruction-, but also density- or distance-based. A benchmark for anomaly segmentation, *i.e.,* anomaly detection methods for semantic segmentation, was recently published in Chan et al. [2021a], providing a cleaner comparison of proposed methods. Given a set of anomalies, the prevailing approach for class discovery is to form clusters based on some similarity measure or intrinsic features with traditional clustering methods. A detailed survey of image clustering has been published in Liu et al. [2021].

**Class-Incremental Learning.**   Class-incremental learning refers to the extension of a neural network's semantic space by further, previously unknown, classes. This extension is achieved by fine-tuning a model on additional, usually human-annotated data [Jung et al., 2018, Li and Hoiem, 2018, Klingner et al., 2020, Michieli and Zanuttigh, 2019], whereas in this work we only provide pseudo labels for these new images. The primary issue to tackle when re-training a neural network is to mitigate the performance loss on pre-

viously learned classes, commonly known as catastrophic forgetting [McCloskey and Cohen, 1989]. To this end, we employ two different strategies: first, we penalize large variations of the softmax output (compared to the one of the original network) [Hinton et al., 2015], second we utilize a subset of the previously-seen training data [Robins, 1995].

The first strategy belongs to the category of regularization based approaches, or more specifically to knowledge distillation methods. These were originally developed to distill knowledge from sophisticated into simpler models [Hinton et al., 2015], *i.e.,* for model compression. Thereupon, distillation methods have evolved for incremental learning in image classification [Li and Hoiem, 2018, Yao et al., 2019, Kim et al., 2019, Jung et al., 2018, Lee et al., 2019], some of which were later adapted to semantic segmentation [Klingner et al., 2020, Michieli and Zanuttigh, 2019, Tasar et al., 2019].

The second approach belongs to so-called rehearsal methods [Robins, 1995], where old training data is included in the re-training process [Rebuffi et al., 2017, Castro et al., 2018].

**Open World.**   The open world setting was first introduced in Bendale and Boult [2015] for image classification. The authors formally define the solution of open world recognition problems as a tuple, consisting of a recognition function, a novelty detector, a labeling process and an incremental learning function. Ideally, these steps should be automated, however, most approaches presume a supervised setting, *i.e.,* they require ground truth for detected novelties. In summary, open world recognition covers the entire process from discovering up to learning novel classes.

A supervised solution for open world object detection is presented in Joseph et al. [2021], based on contrastive clustering, an unknown-aware proposal network and energy based unknown identification. A similar approach was proposed

in Cen et al. [2021] for open world semantic segmentation, where novel classes are learned via few-shot learning. In He and Zhu [2021], an unsupervised method to obtain pseudo labels for image classification based on cluster assignments is introduced. There exists also some prior work for unsupervised open world semantic segmentation [Nakajima et al., 2019], however, the segmentation mask is obtained via agglomerative clustering of superpixels and there is no update of the neural network at all. While it is capable of creating ad hoc novel classes unsupervisedly on given images, it does not create a consistent semantic category over multiple images.

Our work introduces an open world semantic segmentation framework, where a neural network is incrementally extended by novel classes. These classes are discovered **and** labeled without any human effort. Therefore, our work goes beyond all existing approaches in this research area.

## 3 DISCOVERY OF UNKNOWN SEMANTIC CLASSES

Whether a class is novel or not depends on the neural network's underlying set of known classes $\mathcal{C} = \{1, \ldots, C\}$. Let $f : \mathcal{X} \to (0, 1)^{|\mathcal{H}| \times |\mathcal{W}| \times |\mathcal{C}|}$ be a semantic segmentation DNN which is trained on the classes in $\mathcal{C}$, mapping an image $x \in \mathcal{X} \subseteq [0, 1]^{|\mathcal{H}| \times |\mathcal{W}| \times 3}$ onto its softmax probabilities for each pixel $z \in \mathcal{H} \times \mathcal{W}$. Then, $f_{z,c}(x) \in (0, 1)$ denotes the probability with which the model $f$ assigns some pixel $z$ to a class $c \in \mathcal{C}$. As decision rule, we apply the $\arg\max$ function, *i.e.,* we obtain the semantic segmentation mask $m(x) \in \mathcal{C}^{|\mathcal{H}| \times |\mathcal{W}|}$ with $m_z(x) = \arg\max_{c \in \mathcal{C}} f_{z,c}(x)$. In the following, we will estimate the prediction quality on a segment-level instead of pixel-wise, employing a meta regression approach that was first introduced in Rottmann et al. [2020]. On that account, we denote a segment, *i.e.,* a connected component of pixels that share the same class in $m(x)$, as $k \in \mathcal{K}(x)$.

**Meta Regressor.** As model for the meta regressor we apply the gradient boosting from the `scikit-learn v.0.24.2` library using the standard settings. The training datasets contain from 67 to 75 uncertainty metrics depending on the number of classes. We train on $313, 720$ to $946, 318$ segments. Further details on the definition of the segment-wise metrics, the exact size of the training data and the tree models obtained are provided in the Appendix. For any predicted segment $k$, the gradient boosting regressor, via clipping, outputs a value between 0 and 1, where a value close to 0 expresses low, a value close to 1 high prediction quality.

The motivation to use a segment-wise meta regression framework is to identify segments with low predicted IoU as candidate segments that potentially stem from OoD objects.

**Uncertainty Metrics and Prediction Quality Estimation.** We consider novelties as *none-of-the-known* objects, *i.e.,* they differ semantically from the model's training data. Assuming that the segmentation DNN produces unstable predictions on these unexplored entities, various measurable phenomena occur. For instance, the model exhibits a high prediction uncertainty. This is quantified by dispersion measures as the softmax entropy, probability margin or variation ratio, which we compute pixel-wise via

$$E_z(f(x)) = -\frac{1}{\log(|\mathcal{C}|)} \sum_{c \in \mathcal{C}} f_{z,c}(x) \log(f_{z,c}(x)) , \quad (1)$$

$$D_z(f(x)) = 1 - \max_{c \in \mathcal{C}} f_{z,c}(x) + \max_{c \in \mathcal{C} \setminus \{m_z(x)\}} f_{z,c}(x) , \quad (2)$$

$$V_z(f(x)) = 1 - \max_{c \in \mathcal{C}} f_{z,c}(x) , \quad (3)$$

respectively. These are then averaged over the segments $k \in \mathcal{K}(x)$ or over the segment boundary. Moreover, we examine some geometrical properties of the segments, such as their size, *i.e.,* the number of pixels $|k|$ contained in $k$, their shape or their position in the image. For in-depth details on the constructed metrics, we refer to Rottmann et al. [2020] and the appendix. By feeding these metrics into a meta regression model, we obtain prediction quality estimates for each segment $k \in \mathcal{K}(x)$, which we denote by $s(k) \in [0, 1]$. These quality estimates approach the true segment-wise *Intersection over Union* (IoU) with reasonably high accuracy [Rottmann et al., 2020]. To fit the meta regressor, we compute the metrics plus the true IoU values of all segments included in the training data of the segmentation network. This meta model is then applied to unseen data, *i.e.,* data that was not included in the training of $f$, for the purpose of anomaly segmentation. Here, we consider a segment $k$ to be anomalous, if its quality score is below some predefined threshold $\tau \in [0, 1]$, *i.e.,* if $s(k) < \tau$. By that, we identify individual segments as unknown, however, the semantic segmentation of unknown objects usually consists of several segments, *i.e.,* of different predicted classes. As we can uniquely assign each pixel $z$ to a segment $k(z)$, we obtain a binary pixel-wise classification mask $a \in \{0, 1\}^{|\mathcal{H}| \times |\mathcal{W}|}$ via

$$a_z = \mathbb{1}_{\{s(k(z)) < \tau\}} \ \forall z \in \mathcal{H} \times \mathcal{W} , \quad (4)$$

where the class label $\mathbb{1}_{\{s(k(z)) < \tau\}} = 1$ indicates anomalous pixels. Finally, the connected components in the anomaly mask $a$ merge adjacent anomalous segments into suspicious objects. Under ideal conditions,

1. the semantic segmentation network performs perfectly on in-distribution data,

2. the meta model detects all (but only) unknowns, and

3. novel objects of different classes are separable.

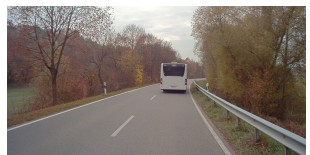 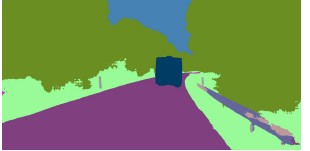 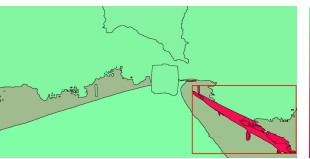 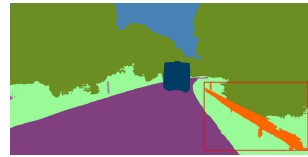

| image from A2D2 | semantic segmentation prediction | prediction quality estimation from 0 (red) to 1 (green) | pseudo ground truth |

Figure 3: Novelty segmentation: example for obtaining pseudo ground truth with regard to some image patch (outlined in red) of image $x$. If segments inside the red box exhibit quality estimates below some predefined threshold, they are "re-labeled" in the segmentation mask $m(x)$.

**Embedding and Clustering of Image Patches.** Image clustering usually takes place in a lower dimensional latent space due to the curse of dimensionality. To this end, we feed image patches tailored to the suspicious objects into an image classification DenseNet201 Huang et al. [2017], which is trained on the ImageNet dataset [Deng et al., 2009] with 1000 classes. The patches are not equally sized. That nevertheless the DenseNet feature extractor returns features of equal size $(1, 920)$ for each patch is a consequence of the application of the AdaptiveAvgPool2d layer that is applied as the last layer after the fully convolutional and depthwise interconnected layers of the DenseNet. Put shortly, this last layer pools over both spatial dimension of the feature maps and thereby the output is not dependent on the size of the input, that is transported through the fully convolutional layers. Their feature representations are further compressed, resulting in a two-dimensional embedding space as illustrated in Fig. 2 (bottom left). We apply two commonly used dimensionality reduction techniques. For complexity reasons, we compute the first 50 principal components [Pearson F.R.S.] before deploying the better performing *t-SNE* method [van der Maaten and Hinton, 2008] with Euclidean distance as similarity measure.

This procedure for image embedding is adopted from Oberdiek et al. [2020], where the authors evaluated several feature extractors, distance metrics and feature dimensions. We employ the best performing setup in this quantitative analysis to obtain clusters of visually related image patches. Beyond that, we identify these clusters using the *DBSCAN* [Ester et al., 1996] algorithm. This clustering method requires two hyperparameters, namely the radius $\varepsilon \in \mathbb{R}$ that defines a neighborhood $B_\varepsilon(\cdot)$ and a threshold $N_{\min} \in \mathbb{N}$ regarding the number of data points within this $\varepsilon$-neighborhood. Let $\mathcal{E} = \{e_1, e_2, \ldots\} \subset \mathbb{R}^2$ denote the set of the embedded features. Then, an embedding is considered a core point, if and only if it has at least $N_{\min}$ neighbors, *i.e.,*

$$e_i \in \mathcal{E} \text{ is core point} \Leftrightarrow$$
$$|\{e_j \in \mathcal{E} : e_j \in B_\varepsilon(e_i)\}| \geq N_{\min} . \quad (5)$$

The algorithm further distinguishes between border points, *i.e.,* embeddings that are not core points themselves, but belong to a core point's neighborhood, and noise else. To mitigate the risk of failures, *i.e.,* objects from a different category in the novel clusters, we only consider the core points. We further reject embeddings representing image patches that are smaller than some predefined size. The cluster with the most remaining core points (or all clusters that involve "enough" core points) will be used to extend the segmentation network by new classes (Fig. 2, bottom).

**Novelty Segmentation.** Using pseudo labels instead of manually annotated targets is a cost-efficient (in the sense of human effort) method of training neural networks on unlabeled data. For the sake of simplicity we assume that exactly one cluster is returned by the aforementioned procedure. For some image $x \in \mathcal{X}$, we denote the predicted segmentation mask by $m(x)$ and the respective segments by $\mathcal{K}(x)$. Let $\mathcal{K}^{\text{novel}}(x) \subseteq \mathcal{K}(x)$ describe the set of segments $k \in \mathcal{K}(x)$ that are also included in the considered cluster. If $\mathcal{K}^{\text{novel}}(x) \neq \emptyset$, *i.e.,* image $x$ (probably) contains the novel class, we include the tuple $(x, \tilde{y}(x)) \in \mathcal{X} \times \{1, \ldots, C+1\}^{|\mathcal{H}| \times |\mathcal{W}|}$ into the re-training data $\mathcal{D}^{C+1}$ for learning the novel class $C + 1$. Here, $\tilde{y}(x)$ denotes the pseudo label, where

$$\tilde{y}_z(x) = \begin{cases} C+1 & \text{, if } k(z) \in \mathcal{K}^{\text{novel}}(x) \\ m_z(x) & \text{, otherwise} \end{cases}, \quad (6)$$

*i.e.,* a pixel $z$ is either assigned to the novel class ID $C+1$, or to the class $c \in \mathcal{C}$ that was predicted by the initial model $f$. An example for acquiring pseudo ground truth for one image is given in Fig. 3. In the following section we extend the segmentation DNN $f$ by fine-tuning it on $\mathcal{D}^{C+1}$.

## 4 EXTENSION OF THE MODEL'S SEMANTIC SPACE

In this section we describe our approach to semantic incremental learning with the pseudo ground truth acquired by novelty segmentation. Starting from our initial segmentation model $f$, we are seeking an extended model $g : \mathcal{X} \to (0, 1)^{|\mathcal{H}| \times |\mathcal{W}| \times (C+1)}$ that retains the knowledge of $f$ while additionally learning the novel class $C + 1$. Denote

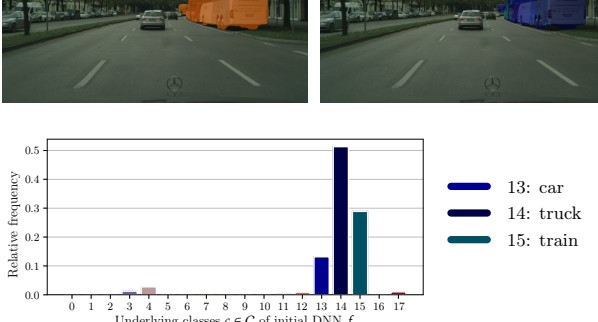

novelty pseudo ground truth    classes predicted by initial DNN

Figure 4: Bar plot showing the relative frequencies of predicted classes for instances of the novel class, together with an exemplary image.

the extended semantic space by $\mathcal{C}^+ = \mathcal{C} \cup \{C+1\}$. In more detail, we replace the ultimate layer of $f$ and reinitialize only the affected weights to obtain the initial model $g$ for re-training, *i.e.*, the model we train on the newly collected data $\mathcal{D}^{C+1}$. As loss function we apply a weighted cross entropy loss [Yi-de et al., 2004], denoted by $l_{\text{ce},\omega}$. The class-wise weights $\omega_c \in (0, 1]$, $c \in \mathcal{C}^+$, are recalculated for each batch based on the inverse class frequency to alleviate class imbalances.

To mitigate the problem of catastrophic forgetting [Mc-Closkey and Cohen, 1989], we pursue two strategies, namely knowledge distillation [Hinton et al., 2015] and rehearsal [Robins, 1995].

Knowledge distillation in class-incremental learning aims at minimizing variations of the softmax output restricted to only the old classes $c \in \mathcal{C}$. This is realized by an additional distillation loss function [Michieli and Zanuttigh, 2021] $l_{\text{d}}$, where

$$
\begin{aligned}
&l_{\text{d}}(g(x), f(x)) \\
&:= -\frac{1}{|\mathcal{H}||\mathcal{W}|} \sum_{z \in \mathcal{H} \times \mathcal{W}} \sum_{c \in \mathcal{C}} f_{z,c}(x) \log(g_{z,c}(x)) \,.
\end{aligned} \tag{7}
$$

Overall, we aim at minimizing the objective

$$
\begin{aligned}
L := \ &\lambda \, \mathbb{E}[l_{\text{ce},\omega}(g(x), \tilde{y}(x))] \\
&+ (1-\lambda) \, \mathbb{E}[l_{\text{d}}(g(x), f(x))], \ \ \lambda \in [0, 1]
\end{aligned} \tag{8}
$$

with $\lambda$ regulating the impact of the distillation loss.

Rehearsal methods propose to replay (some of) the data $\mathcal{D}^{\text{train}} \subset \mathcal{X} \times \mathcal{C}^{|\mathcal{H}| \times |\mathcal{W}|}$ seen during the training of the initial model $f$. We select a subset $\mathcal{D}^{\text{known}} \subseteq \mathcal{D}^{\text{train}}$ that contains as much data as $\mathcal{D}^{C+1}$. This subset is chosen largely at random, but in such a way that it involves classes, that are

1. not or rarely present in $\mathcal{D}^{C+1}$ (class frequency), or

2. similar or related to the novel class.

As there is no measure for the second case, we identify those classes by considering the frequency, with which a class is predicted by $f$ on pixels assigned to the novel class. This is, for all data $(x, \tilde{y}(x)) \in \mathcal{D}^{C+1}$ and classes $c \in \mathcal{C}$, we sum up the number of pixels $z \in \mathcal{H} \times \mathcal{W}$ where $\tilde{y}_z(x) = C+1 \wedge m_z(x) = c$. An example is given in Fig. 4, where the classes *truck*, *train* and *car* are the most frequently predicted classes for instances of the novel class *bus*.

## 5 EXPERIMENTAL SETUP & EVALUATION

We evaluate our approach on the task of detecting and incrementally learning novel classes in traffic scenes, for which there exist large datasets such as Cityscapes [Cordts et al., 2016] and A2D2 [Geyer et al., 2020]. To this end, all evaluated segmentation DNN's were trained on a training split and only on a subset of all available classes. We then perform our experiments on a test split of the same dataset on which the DNN was trained in order to extent it by exactly one or even multiple novel classes. We measure the performance of the extended models computing the evaluation metrics *intersection over union* (IoU), *precision* and *recall* for a validation set.

**Experimental Setup.** As segmentation DNNs we employ the DeepLabV3+ [Chen et al., 2018b] and the PSPNet [Zhao et al., 2017]. The first is trained for different subsets of known classes on the Cityscapes dataset. Moreover, both models are pre-trained on Cityscapes with all 19 classes and then fine-tuned on the A2D2 dataset. Here we use a label mapping between both datasets through which 14 classes remain.

We perform five experiments: For the first three experiments, a DeepLabv3+ with a WideResNet38 backbone is trained on the Cityscapes dataset, where 1) the classes *person* & *rider*, 2) the class *bus* and 3) the classes *person* & *rider*, *bus* and *car* are excluded. In a fourth experiment, a DeepLabv3+ as well as a PSPNet based on a ResNet50 backbone are fine-tuned on the A2D2 dataset, for which we specified subsets for training, testing and validation, including 2975, 1355 and 451 annotated images, respectively. Then, we also apply our method to the A2D2 dataset without prior fine-tuning, *i.e.*, under a domain shift, employing a DeepLabV3+ trained on Cityscapes. Our experiments follow a hierarchical structure with increasing complexity:

1. Construction of a "well" separated category (*human*),

2. Construction of a category in the midst of known similar categories (*bus*),

3. Construction of multiple novel categories (*human*, *bus*

and *car*),

4. Construction of a new category under domain shift with ground truth for known classes (*guardrail*, with fine-tuning),

5. Construction of a new category under domain shift without ground truth (*guardrail*, without fine-tuning).

Each of those initial DNNs is employed to predict the semantic segmentation masks for the images contained in the respective test set. For the segment-wise prediction quality estimation introduced in Sec. 3, we apply a gradient boosting model to obtain the quality scores $s(k) \in [0, 1]$ for each segment $k \in \mathcal{K}(x)$ and image $x$ in the test set. The threshold in Eq. (4) is set to $\tau = 0.5$, *i.e.*, a segment $k \in \mathcal{K}$ is considered as anomalous, if $s(k) < 0.5$. To extract features of the suspicious objects, we employ a DenseNet201 [Huang et al., 2017], trained on the ImageNet dataset [Deng et al., 2009] with 1000 classes. Note that the DBSCAN hyperparameters have to be selected dependent on the density of the desired clusters.

For the class-incremental extension of an initial DNN $f$, we replace its final layer to obtain a larger DNN $g$ (see Sec. 4). Only the decoder of this model is trained for 70 epochs on the newly collected data $\mathcal{D}^{C+1}$ together with the replayed data $\mathcal{D}^{\text{known}}$. We use random crops of size $1000 \times 1000$ pixels, the Adam optimizer with a learning rate of $5 \cdot 10^{-5}$ and a weight decay of $10^{-4}$. Further, the learning rate is adjusted after every iteration via a polynomial learning rate policy [Chen et al., 2018a]. The distillation loss and the cross-entropy loss are weighted equally in the overall loss function defined in Eq. (8), *i.e.*, $\lambda = 0.5$ (analogously to Michieli and Zanuttigh [2019]).

As the five experiments struggle with different issues, the experimental setup slightly differs. For the first case, we construct the novel category *human*, which is "well" separable from all known classes, to enhance the purity of the "human cluster" and to simplify the learning of novel objects. However, we observe that the DNN tends to "overlook" many humans, *i.e.*, they are assigned to the class predicted in the background, *e.g.*, to the *road* class. As a consequence, the segment-wise anomaly detection fails to detect such persons, which is why these will be assigned to other classes in our acquired pseudo ground truth. To not distract the extended segmentation network, we modify the pseudo labels by ignoring all known classes $c \in \mathcal{C}$ during the incremental training procedure. The *bus* class added in the second experiment is closely related to other classes in the vehicle category, such as *truck*, *train* and *car*, which complicates the construction of pure clusters. We mitigate the impact of objects from similar classes by discarding all objects from the cluster that consist of only one segment in the predicted segmentation. Experiment three extends the previous ones by facing multiple unknown classes, namely *human*, *bus* and *car*. The last two experiments deal with an addi-

| Model | mIoU$_\mathcal{C}$ | IoU$_{\text{novelty}}$ | mIoU$_{\mathcal{C}+}$ |
|---|---|---|---|
| **1. experiment:** Cityscapes, human | DeepLabV3+ | | |
| initial DNN | 68.63 | 00.00 | 64.82 |
| extended DNN (ours) | 68.53 | 39.80 | 66.94 |
| extended DNN (supervised) | 69.43 | 59.33 | 68.87 |
| oracle | 71.05 | 72.85 | 71.15 |
| **2. experiment:** Cityscapes, bus | DeepLabV3+ | | |
| initial DNN | 66.94 | 00.00 | 63.42 |
| extended DNN (ours) | 67.07 | 44.73 | 65.89 |
| extended DNN (supervised) | 66.74 | 41.40 | 65.41 |
| oracle | 69.48 | 76.66 | 69.86 |
| **3. experiment:** Cityscapes, multi | DeepLabV3+ | | |
| initial DNN | 56.99 | 00.00 & 00.00 | 50.29 |
| extended DNN (ours) | 57.52 | 40.22 & 81.27 | 57.90 |
| oracle | 77.28 | 81.90 & 94.94 | 78.59 |
| **4. experiment (a):** A2D2, guardrail | DeepLabV3+ (fine-tuned) | | |
| initial DNN | 75.77 | 00.00 | 70.72 |
| extended DNN (ours) | 72.07 | 46.10 | 70.34 |
| oracle | 75.23 | 74.58 | 75.19 |
| **4. experiment (b):** A2D2, guardrail | PSPNet (fine-tuned) | | |
| initial DNN | 68.77 | 00.00 | 64.19 |
| extended DNN (ours) | 64.54 | 32.79 | 62.42 |
| oracle | 67.71 | 69.08 | 67.80 |
| **5. experiment:** A2D2, guardrail | DeepLabV3+ (not fine-tuned) | | |
| initial DNN | 59.38 | 00.00 | 55.42 |
| extended DNN (ours) | 60.48 | 20.90 | 57.84 |

Table 1: Comparing overview of all evaluated models, where the results for our extended DNNs are highlighted in gray. As performance metrics, we provide the mean IoU over the old and new classes, denoted by mIoU$_\mathcal{C}$ and mIoU$_{\mathcal{C}+}$, respectively, and the IoU value of the novel class(es), IoU$_{\text{novelty}}$.

tional domain shift from urban street scenes in Cityscapes to countryside and highway scenes in A2D2. To bridge this gap, we fine-tune the initial DNN on our A2D2 training set, which, however, requires A2D2 ground truth for the known classes. Without fine-tuning, the prediction quality and thereby the quality of our pseudo ground truth suffers. On that account, we discard images that are generally rated as badly predicted, *i.e.*, where the relative amount of pixels with a low quality estimate exceeds $1/3$ of the image in total. Moreover, we renounce the replay of previously-seen data, since this prevents the DNN from adapting to the new domain.

**Evaluation of Results.** In the following, all evaluation values belonging to our extended models are averaged over five runs of the respective experiment. For in-depth details we refer to the appendix. We provide a qualitative comparison of different models for all conducted experiments in Tab. 1, reporting the mean IoU over the known classes and over the extended class set, denoted as mIoU$_\mathcal{C}$ and mIoU$_{\mathcal{C}+}$, respectively, as well as the IoU value of the novel classes (IoU$_{\text{novelty}}$). The models considered in this comparison are the initial and the extended DNN, where the class space is extended via our method. For the first and second experiment we further compare our approach with a baseline, where a DNN is extended using a self-training approach. That is, we employ a so-called teacher network, which is al-

| | IoU | precision | recall | IoU | precision | recall |
|---|---|---|---|---|---|---|
| **1. experiment:** | | | DeepLabV3+ | | | |
| Cityscapes, human | | initial | | | extended | |
| human | 00.00 | 00.00 | 00.00 | 39.80 | 60.60 | 53.72 |
| mean over $\mathcal{C}$ | 68.63 | 79.79 | 80.94 | 68.53 | 83.32 | 77.17 |
| mean over $\mathcal{C}^+$ | 64.82 | 75.36 | 76.44 | 66.94 | 82.05 | 75.86 |
| **2. experiment:** | | | DeepLabV3+ | | | |
| Cityscapes, bus | | initial | | | extended | |
| bus | 00.00 | 00.00 | 00.00 | 44.73 | 58.33 | 66.15 |
| mean over $\mathcal{C}$ | 66.94 | 79.32 | 79.55 | 67.07 | 82.46 | 76.31 |
| mean over $\mathcal{C}^+$ | 63.42 | 75.15 | 75.36 | 65.89 | 81.19 | 75.78 |
| **3. experiment:** | | | DeepLabV3+ | | | |
| Cityscapes, multi | | initial | | | extended | |
| human | 00.00 | 00.00 | 00.00 | 40.22 | 68.74 | 49.65 |
| car | 00.00 | 00.00 | 00.00 | 81.27 | 86.56 | 93.05 |
| mean over $\mathcal{C}$ | 56.99 | 65.75 | 80.88 | 57.52 | 78.53 | 65.77 |
| mean over $\mathcal{C}^+$ | 50.29 | 58.01 | 71.37 | 57.90 | 78.43 | 66.43 |
| **4. experiment (a):** | | | DeepLabV3+ | | | |
| A2D2, guardrail | | initial | | | extended | |
| guardrail | 00.00 | 00.00 | 00.00 | 46.10 | 80.41 | 52.09 |
| mean over $\mathcal{C}$ | 75.77 | 87.86 | 83.47 | 72.07 | 89.01 | 78.44 |
| mean over $\mathcal{C}^+$ | 70.72 | 82.00 | 77.90 | 70.34 | 88.44 | 76.69 |
| **4. experiment (b):** | | | PSPNet | | | |
| A2D2, guardrail | | initial | | | extended | |
| guardrail | 00.00 | 00.00 | 00.00 | 32.79 | 70.75 | 38.04 |
| mean over $\mathcal{C}$ | 68.77 | 84.57 | 76.79 | 64.54 | 86.41 | 71.22 |
| mean over $\mathcal{C}^+$ | 64.19 | 78.93 | 71.67 | 62.42 | 85.36 | 69.01 |
| **5. experiment:** | | | DeepLabV3+ | | | |
| A2D2, guardrail | | initial | | | extended | |
| guardrail | 00.00 | 00.00 | 00.00 | 20.90 | 77.12 | 22.32 |
| mean over $\mathcal{C}$ | 59.38 | 79.50 | 68.14 | 60.48 | 84.08 | 66.61 |
| mean over $\mathcal{C}^+$ | 55.42 | 74.20 | 63.60 | 57.84 | 83.61 | 63.66 |

Table 2: Direct comparison of the initial and the extended DNNs for all conducted experiments. We report the IoU, precision and recall values for the novel class (highlighted with gray rows), respectively, as well as averaged over the previously-known and the extended class spaces $\mathcal{C}$ and $\mathcal{C}^+$.

ready trained on the extended semantic space $\mathcal{C}^+$, to produce pseudo labels for some student network. Thereby, we obtain a high quality pseudo ground truth. Apart from this, the baseline DNN is extended analogously to ours. In addition, for the first four experiments we provide results of an *oracle*, *i.e.,* a DNN, that is initially trained on the extended class set $\mathcal{C}^+$ and only with human-annotated ground truth. In the fifth experiment, we extend the initial DNN by a novel class derived from a different dataset. To some extent, the oracle from experiment four (a) can serve as a coarse reference for experiment five. In Tab. 2 we give a more detailed overview about all experiments, reporting not only the IoU, but also the precision and recall values of the novel class as well as averaged over $\mathcal{C}$ and $\mathcal{C}^+$. Note that the fourth experiment is evaluated twice, once for (a) the DeepLabV3+ and once for (b) the PSPNet. For class-wise evaluation results and visualizations, we refer to Appendix A.

In general, we observe that our approach succeeds in incrementally extending a DNN by a novel class, while the performance on previously-known classes remains stable. On Cityscapes, we achieve IoU values for the novel classes human and bus of $\text{IoU}_{\text{human}} = 39.80 \pm 0.73\%$ and

$\text{IoU}_{\text{bus}} = 44.73 \pm 1.46\%$, respectively. For the third experiment with two novel classes, we obtain similar results for the *human* class with $\text{IoU}_{\text{human}} = 40.22 \pm 1.77\%$ and for the *car* class even $\text{IoU}_{\text{car}} = 81.27 \pm 1.16\%$. While these IoU values are a considerable achievement for a method working without ground truth, the distinct gaps to the oracle's IoU values still leave room for further improvement. Compared to the baseline DNN, we do not achieve competitive performance in the first experiment, while in the second experiment, our approach actually performs slightly better. This is explained by the fact, that the pseudo ground truth for the *human* class incorporates much more noise than that for the *bus* class. In the fourth experiment we mitigate the domain shift from Cityscapes to A2D2 by prior fine-tuning of the networks, using A2D2 ground truth. By that, we obtain IoU values of $\text{IoU}_{\text{guardrail}} = 46.10 \pm 4.8\%$ for the DeepLabV3+ and $\text{IoU}_{\text{guardrail}} = 32.79 \pm 3.48\%$ for the PSPNet. We conclude, that our approach achieves better results for models which are initially better-performing. Without fine-tuning the DeepLabV3+ on A2D2, we obtain $\text{IoU}_{\text{guardrail}} = 20.90 \pm 1.73\%$, while the mean IoU over the previously-known classes $\mathcal{C}$ slightly increases from $59.38\%$ to $60.48 \pm 0.47\%$.

## 6 CONCLUSION & OUTLOOK

In this work, we have introduced a new and modular procedure for the class-incremental extension of a semantic segmentation network, where novel classes are detected, annotated and learned in an unsupervised fashion. While there already exists an unsupervised open world approach for semantic segmentation [Nakajima et al., 2019], we are the first in this field to extend a neural network's semantic space by robust novel classes. We performed five hierarchically structured experiments with an increasing level of difficulty. We demonstrated that our approach can deal with novelties that are either "well" separated or related to known categories, and that it is even applicable when the test data is sampled from a slightly different distribution than the DNN was trained on. Moreover, we applied two different models in the fourth experiment, where the initial DeepLabV3+ already outperformed the initial PSPNet. This performance gap is also reflected in the model's ability to learn the novel class, thus we conclude that our method benefits significantly from high performance networks.

For future work, we plan to improve the extension of a neural network by multiple classes at once. On that account, suitable datasets are in demand. Two datasets for the task of anomaly segmentation were recently published in Chan et al. [2021a], however, these show a wide variety of anomalous objects. To advance the research in class-incremental learning, it requires datasets where novel objects, *i.e.,* objects that do not appear in the training data, appear frequently in the test data.

We are currently working on a synthetic dataset tailored to our approach. This data is generated using the CARLA 0.9.12 simulator Dosovitskiy et al. [2017], similar as extensively described in Kowol et al. [2022]. The data include annotated street scene images, generated on the same maps for training and testing. Since we aim at detecting novel classes in the test data, these images are enriched by several **never-seen** object classes, *e.g., deer*, *construction vehicle* or *portable toilet* (examples provided in Appendix B).

Besides, we plan to adapt our approach to video instead of image data, where anomaly detection includes anomaly tracking over multiple frames.

Our source code is publicly available on github under https://github.com/SUhlemeyer/novelty-learning.

# 7 LIMITATIONS & NEGATIVE IMPACT

With the procedure presented in this work, we are taking a first step towards a new machine learning problem. This first step is highly experimental and our method has not the technology readiness level to be applied to real-world problems in a fully automated fashion. Especially from the safety point of view, a neural network should not be modified without any supervision, since we can not guarantee to avoid significant performance drops.

### Acknowledgements

This work is funded by the German Federal Ministry for Economic Affairs and Energy, within the project "KI Delta Learning", grant no. 19A19013Q. We thank the consortium for the successful cooperation. The authors gratefully also acknowledge the Gauss Centre for Supercomputing e.V. (https://www.gausscentre.eu) for funding this project by providing computing time through the John von Neumann Institute for Computing (NIC) on the GCS Supercomputer JUWELS at Jülich Supercomputing Centre (JSC).

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
