# OpenReview forum: "Towards Unsupervised Open World Semantic Segmentation"
_auai.org/UAI/2022/Conference — UAI 2022 Poster_

### Official Review · Reviewer_k9CT · 2022-04-10

**Q2(1) Originality/Novelty:** 2
**Q2(2) Significance/Impact:** 3
**Q2(3) Correctness/Technical Quality:** 2
**Q2(6) Clarity Of Writing:** 3
**Q6 Overall Score:** 5
**Q8 Confidence In Your Score:** 3

**Q1 Summary And Contributions:**

The paper proposed a new and modular approach to the problem of open-world semantic segmentation. The method leverages the fact that unknown objects are clustered in low-dimensional feature spaces. Those clusters are then utilised to define new classes and serve as training data for unsupervised incremental learning. Authors claim the work is the first in the field to extend a network’s semantic space by robust novel classes.


**Q2 Assessment Of The Paper:**

More detailed information regarding each of these aspects is given below:

**Q2(4) Quality Of Experiments (Optional):**

3: Good: The experimental evaluation is adequate, and the results convincingly support the main claims.

**Q2(5) Reproducibility:**

2: Fair: Key resources (e.g., proofs, code, data) are unavailable but key details (e.g., proof sketches, experimental setup) are sufficiently well-described for an expert to confidently reproduce the main results.

**Q3 Main Strengths:**

1. Experiments regarding the accuracy of the proposed method is sufficient.
2. The paper is in general carefully written and clear to follow.
3. It is good to have a disclaimer in Section 7 on the applications with safety concerns.


**Q4 Main Weakness:**

1. There are several components in the proposed framework, it is unclear about how the performance of individual components (e.g., the embedding network in Fig. 14) affects the overall results. Sensitivity analysis in this regard may strengthen the paper.
2. Apart from the accuracy experiments that have been done, it would be great to study the robustness of the proposed methods, e.g., with perturbations on images.


**Q5 Detailed Comments To The Authors:**

The paper is targeting an interesting topic and introduced a simple yet effective method. It is highly modularized with low coupling, and each of the 4 main steps is built upon some 3rd party techniques. I wonder if those technical solutions can be replaced by others, and how the overall performance depends on individual steps (cf. main weakness point 1). In the second step, why reduce the dimensionality of features up to only two? Please be clearer on the choice of the number of dimensionalities.

Minors:
1. “…, were novel classes are detected,” -> “…, where novel…”
2. The quite useful figure 14 should be relocated into the main paper.

**Q7 Justification For Your Score:**

The main strengths point 1 and 2 are weighed most heavily in my assessment.

**Q9 Complying With Reviewing Instructions:**

1: Yes.

---

### Official Review · Reviewer_DGfo · 2022-04-12

**Q2(1) Originality/Novelty:** 3
**Q2(2) Significance/Impact:** 3
**Q2(3) Correctness/Technical Quality:** 3
**Q2(6) Clarity Of Writing:** 3
**Q6 Overall Score:** 7
**Q8 Confidence In Your Score:** 4

**Q1 Summary And Contributions:**

The paper proposes a method to perform unsupervised incremental learning in an open world where new classes might appear by 1) assess quality of the segments in the prediction map via a meta regressor, 2) the segments with low estimated quality are clustered based on “visual similarity”, 3)  generate psuedolabels from clusters , and finally 4)  retraining the network using the psuedolables.

The main novelty is using the cluster group of 2d features to generate pseudolabels for retraining.


**Q10 Ethical Concerns (Optional):**

I have no ethical concerns.

**Q2 Assessment Of The Paper:**

More detailed information regarding each of these aspects is given below:

**Q2(4) Quality Of Experiments (Optional):**

3: Good: The experimental evaluation is adequate, and the results convincingly support the main claims.

**Q2(5) Reproducibility:**

3: Good: Key resources (e.g., proofs, code, data) are available and key details (e.g., proofs, experimental setup) are sufficiently well-described for competent researchers to confidently reproduce the main results.

**Q3 Main Strengths:**

Little is done on this hard problem and the use of pseudolabels is clearly a good idea.

The paper is well-written and has a nice structure.

It is mentioned that datasets for this type of problems do not yet exist, but the experiments on existing datasets show that it is not needed to develop an algorithm that performs this type of task.


**Q4 Main Weakness:**

The paper combines several know techniques, and it could be argued that the difference between what is found in the literature and the novel ideas presented in this paper is small.

The experiments are restricted. They only show cases where one, two or three new classes are added. The real-world is more complex than this.

Results from only one experiment run for each of the five experiments is reported.

The data that is used for the experiments are open, but the exact versions of these that is used are not shared. These should be shared with the code.

**Q5 Detailed Comments To The Authors:**

This is a nice paper that solves a little researched problem. It is well-written and has a good structure. The paper provides the right level of detail, especially if code is shared as is mentioned in the conclusion. I found this paper to be a good read.

I think that the novelty is good enough. The combination of known techniques is not straight forward and the new idea (with using pseudolabels) locks up the proposed solution. I would argue that having a paper the presents a solution for unsupervised segmentation on new classes could spur new and maybe more novel ideas.

The experiments shows that it generates good results without any human efforts (mIoU of ~40% compared to 60-70% for known classes). However, it is a small set of new classes. It would be interesting to see how it works on an increasing number of classes. How will the performance degrade (will it?) if an increasing number of classes are introduced? The experiments do not show negative examples. When does the solution fail? What is its limit.

The papers do not discuss limitations of the research. Where will it fail? What are the assumptions and when are they not met?

It looks like experiments are only run once. Hence, central tendencies (mean) and variation are not reported. Thus, it is not clear whether the findings are significant. Some of them clearly are, but others are close.

This is important as there are many sources of variation in this paper, both related to algorithmic factors and implementation factors. It is not clear how robust the method is to these factors [1,2]. This is a problem with lots of the published results in machine learning, and the lack of multiple runs hides many false findings.

Single run of all experiments is my main issue with the paper is the. The main machine learning conferences do not accept papers with only one run, and there are reasons for this. Please add more runs with mean and variation.

The redeeming quality for exactly this issue is the novelty of the problem solved. Hence, the main point is not how the algorithm compares to other methods, but that you are able to do this in the first place. However, for those that will build on your work, it will be much harder to understand whether their methods are improvements when variation is not reported, which might lead to false claims of improvement over this method.

It is not clear how hyperparameter optimization is done, and whether the baseline is performing at anything close to state of the art (SotA). If not, the results are less interesting. It is not discussed at all how the baseline compares to SotA.
Overview of method in 1 introduction is very nice. Please make an explicit connection between these steps and what is described in sections 3 and 4. This will improve readability quite a lot.

There is a bit confusing terminology in the paper. For example, the abstract uses the term connected components, which could either mean segments or segmentation mask or image patch or suspicious objects or novel objects. Please standardize to fewer terms that are always used in the text. It would be nice if these terms were clearly defined as well. Furthermore, segmentation mask is a term that has connotations that are different from how I understand it used in the paper. Cleaning up the use of these terms and reducing them will help readability quite a lot and make the paper even easier to read. Fixing these issues would change the paper from the current "good" to an  "excellent" score for quality of writing.

It is not clear how image patches with different dimensions are fed to the image classification network that is used for feature extraction. Please elaborate using one (?) sentences. A lot of sentences are not required, but it would be nice to understand how this is done. A pointer to a reference or two would be the least requirement.

It is not clear how you find the “best performing setup” (line two, page five). Did you just pick it from the referenced paper, or did you run own experiments? Please elaborate.

I find the section on novelty segementation hard to understand, especially the realtion to the segmentation mask m(x). Also, equation 6 is not crystal clear. It is the part with the segmentation mask m_z(x) that is not clear to me. Please improve the description.

Thanks for adding the appendix with details on the performance. Much appreciated.

Releasing the code will help with all the details that there is no space for in a conference paper. The method you present relies on many moving parts and the code will help smooth out the understanding of details for someone who would like to build on you work. Thanks for sharing. Much appreciated.

The data that is used for the experiments are publicly available, but the exact versions of these that is used are not shared. These should be shared with the code. By sharing the actual sets used, my score on reproducibility would be increased to "excellent".

There are some inconsistencies among results of table1 and 2. For example, the mIoU of the extended model on C+ is 66.52 in table 1 and 66.75 in table2. Please explain why it is so.

Some detailed comments:
* page 5 second paragraph: “and noise else”
* page 6, section 5. it is not clearly stated that only for experiment 4 and 5 models are pre-trained on all 19 classes which lead to confusion.
* page 7, right column, first paragraph. it is required to have a dot before “In the fifth experiment”.

[1] PHAM, Hung Viet, et al. Problems and opportunities in training deep learning software systems: An analysis of variance. In: Proceedings of the 35th IEEE/ACM international conference on automated software engineering. 2020. p. 771-783.

[2] ZHUANG, Donglin, et al. Randomness in neural network training: Characterizing the impact of tooling. arXiv preprint arXiv:2106.11872, 2021.


**Q7 Justification For Your Score:**

I like this paper a lot. It contains interesting solutions to a hard problem that needs more investigation. This paper could spawn more research on open world segmentation, which I think would be a good thing.

My main complaint is the single run of all experiments.

The only reason for me not rejecting based on this issue alone it the novelty of the problem solved. The main point is not how the algorithm compares to other methods, but that you are able to do this.


**Q9 Complying With Reviewing Instructions:**

1: Yes.

---

### Official Review · Reviewer_7Nqy · 2022-04-12

**Q2(1) Originality/Novelty:** 3
**Q2(2) Significance/Impact:** 2
**Q2(3) Correctness/Technical Quality:** 3
**Q2(6) Clarity Of Writing:** 2
**Q6 Overall Score:** 6
**Q8 Confidence In Your Score:** 5

**Q1 Summary And Contributions:**

This paper presents a method that allows a pre-trained semantic segmentation network to discover new classes and learn to segment them. In other words, the work presents a method for unsupervised class-incremental learning, since there is no target label supervision for the new classes. The main contribution is that this paper (to the best of my knowledge) is the first to present a method for this particular setup. The results show reasonably strong IoUs, given that no ground truth exists.

**Q2 Assessment Of The Paper:**

More detailed information regarding each of these aspects is given below:

**Q2(4) Quality Of Experiments (Optional):**

3: Good: The experimental evaluation is adequate, and the results convincingly support the main claims.

**Q2(5) Reproducibility:**

2: Fair: Key resources (e.g., proofs, code, data) are unavailable but key details (e.g., proof sketches, experimental setup) are sufficiently well-described for an expert to confidently reproduce the main results.

**Q3 Main Strengths:**

The main strength is that this paper presents a new problem setup: discovering new categories and giving an already trained model the ability to extend its class vocabulary "on the fly". An illustrative example would be a semantic segmenter for self-driving cars, that sees an e-scooter for the first time and needs to quickly adapt to recognize all other e-scooters as well, after they were introduced to their city overnight.
The problem is not well studied, because it is difficult and ambiguous to evaluate (i.e., human annotators would disagree on whether or not to label something a new category or not). However, it is important to establish baselines for this problem, which is why this paper could be a valuable addition to the literature.

**Q4 Main Weakness:**

The main weakness is the clarity of the presentation. In the following, I list some representative examples that lead to a decrease in clarity:

1. There is no figure in the main paper that shows the overall flow of the method. Figure 14 on the last page of the appendix shows the method, but it should be in the main paper. The method figure is what a typical reader looks at first, and it helps immensely in understanding the content.

2. Since the paper introduces a new training setup, it would be better to clarify in detail how the dataset is prepared early in the paper (both in intro and method), i.e. by giving the example of leaving out the bus and person class, as desribed later in the experiments section.

3. Important details are omitted in the textual description of the method. One example is the meta-regression model: Even if the details can be read in (Rottman et al. 2020), the paper itself should be self-contained.

(for details, see detailed comment section)

**Q5 Detailed Comments To The Authors:**


1. Figure 14 should be put in the main paper and be given a descriptive caption that explains the model in broad strokes.

2. It would be good to add a few sentences in the intro about the training and evaluation setup. How is the dataset splitted? How many unknown classes are used?

3. It would be good to explain the colors in the "predicition quality estimation" panel of Figure 3 in the figure caption, or by adding a color bar.

4. What network architecture does the meta regressor have? What is the input and output dimension, and what is the network type? What are the "constructed metrics"? It seems like every segment is summarized by a feature vector that contains heuristic metrics?

5. "To fit the meta regressor, we compute the metrics plus the true IoU values of all segments". This sentence may confuse readers not familiar with (Rottmann 2020). Is the regressor a network that is trained to predict the IoU value from the metrics? If yes, what is the motivation?

6. "To this end, we feed image patches tailored to the anomalies into an image classification CNN". How exactly do you feed the patches into the CNN? Do you cut out a rectangle around the irregular shapes, and then upscale it to a uniform image size? Or do multiply the image with the anomaly mask and feed the result into the CNN? Further, does the CNN give you a single vector for each segment, using some pooling operation, or does it return feature maps? What is the dimension of the features?

7. Concerning DBSCAN, why do you cluster in 2 dimensions, and not let say 3 or more? Does an ablation exist?

8. In the paragraph on Embeddings at the end of page 4, what does "[F.R.S."]" stand for?

9. "We further reject embeddings representing image patches that are smaller than some predefined size" - how do you predefine the size? A negative point of the algorithm seems to be that there are a lot tiny bells and whistles that have to be tuned by hand and adapted to every dataset, or to which class on may like to detect. Other examples include: modifying known classes during incremental learning, disregarding objects from clusters that consist of only one segment in the prediction, and the threshold of 33% low quality estimates that lead to reject a training example.

10. The clustering algorithm returns the points (segments) that belong to the best defined unknown cluster. What would happen if the algorithm extends the set of classes C by more than one? (i.e. C+2 instead of C+1).

11. The gradient boosting is not further explained. A short explanation of what exactly is done there and why should be added. The paper should be reproducible from reading the main paper + appendix. Things that are not explained in the main paper should have references to the appendix section that explains them.

12. Please explain the A2D2 dataset in more detail, especially how many classes it has and how much overlap there is to in terms of classes and scenery with Cityscapes.

13. Why is the guardrail class picked out of all classes? There should be an explanation. What happens if we pick other classes?

14. Could you explain the limits in more detail. For example, could one learn all 19 Cityscapes classes in sequence?

**Q7 Justification For Your Score:**

The paper introduces an important problem (very good) but lacks in clarity (not so good), averaging to a borderline case.

**Q9 Complying With Reviewing Instructions:**

1: Yes.

---

### Decision · Program_Chairs · 2022-05-15

**Decision:**

Accept (Poster)

**Comment:**

Meta Review: The paper proposes a method that allows a pre-trained semantic segmentation network to discover new classes and learn to segment them in an open world. Two reviewers appreciate the key idea and acknowledge that this is an important problem. Reviewer 3 raises concerns about the lack of sensitivity analysis. In all, the meta-reviewer considers the paper to be a valuable contribution. The authors need to incorporate the reviews when preparing the camera-ready version.